# A University Foreign Language Curriculum for Pre-Service Non-Language Subject Teacher Education

**Elena Borzova \* and Maria Shemanaeva \***

Institute of Foreign Languages, Department of the English Language, Petrozavodsk State University, 185910 Petrozavodsk, Russia

**\*** Correspondence: anat.bor@onego.ru (E.B.); indy2002@mail.ru (M.S.)

**Abstract:** The article focuses on the problems of teaching foreign languages (FL) to pre-service subject (other than foreign languages) teachers and highlights their possible solutions. A sustained discontent of both university teachers and students with the outcomes of FL classes gave impetus for a systematic, detailed study of the university FL education of the target group of learners. There was an acute need to develop well-grounded guidelines for the related course objectives, its content and adequate techniques intended for the target group of learners. The authors claim that the generally accepted English for Specific Purposes (ESP) or Content and Language Integrated Learning (CLIL) models of university FL education cannot be applied in this context for a number of reasons. A limited amount of classroom time, lack of updated teaching materials, mixed-ability classrooms, and unclear perspectives of the future application of the FL skills made the study urgent. Therefore, the teachers of the English Language Department of the Petrozavodsk State University (Russia) developed a project that is described in the article; 152 students and six teachers participated in its implementation. The objective of the project was to introduce reasonable changes into the approaches, the aims, the content, and the techniques for rational and effective FL teaching and learning in this context. The authors identified priorities of the university FL education in line with the context and the target learner characteristics. The article presents arguments in favor of the introduced changes, describes the phases of the project work, and gives details of all the constituents of the proposed curriculum. Based on the outcomes of the project, the authors conclude that it is necessary to undertake further research steps in this direction.

**Keywords:** university foreign language education; bachelor's degree non-language teacher training

---

## 1. Introduction

University FL departments teach languages to pre-service subject teachers for whom foreign languages are not a major. The peculiarity of the FL education of this group of learners is that it hardly falls into any currently used models of teaching FLs for specific purposes. Moreover, the teaching and learning conditions of this group of learners are far from being ideal in the majority of Russian universities. An unjustifiably short course (first year of university studies), a limited number of classroom time, as well as mixed-ability student groups make the achievement of the objectives defined by the Federal State University Education Standards (FSUES) in this area problematic. As a result, both teachers and students are rarely satisfied with the outcomes. The purpose of this article is to highlight the most critical issues of FL education in this context and suggest a few ways of how to change the curriculum. The staff of the English Language Department of the Petrozavodsk State University (Russia) launched a project to investigate the conditions of the university FL education of the target group of students and to find well-grounded solutions for the problems that arose. The general research questions that we posed in the project were, "What changes are to be introduced into the FL

curriculum intended for university pre-service subject teachers to provide acceptable FL proficiency levels of the target group of learners? What are the guidelines for implementing the proposed changes in university FL teaching and learning?"

The project implemented by the FL staff of the PetrSU had three main objectives:

- To map out and introduce reasonable changes into the FL university curriculum intended for pre-service non-language subject teachers in terms of the achievable teaching and learning goals and content;
- To develop effective as well as easily applicable techniques for FL teaching and learning that will allow students to update and further develop their FL competency in the future;
- To boost students' positive motivation for FL learning.

The expected outcome of the project was a restructured curriculum of teaching FLs to the target group of university students.

The basic assumption underlying the research was that the low FL proficiency level of the overwhelming majority of the first-year non-language students and the limited time allocated to learning an FL made it crucial to introduce changes into the curriculum in order to adjust the objectives, the content, and the techniques to the reality of university FL education of the target group of learners. The changes to be made consisted in setting realistic objectives, selecting realistic content, and using efficient techniques to provide an acceptable FL proficiency level for every learner.

## 2. Participants

### 2.1. Students

The target student participants of the project were students in non-language pre-service teacher education who studied for a bachelor's degree to become subject teachers [physical education (PE), biology, history, etc.]. On the whole, there were 152 students who did the placement test, attended classes, and did the final test at the end of the project. The project was completed in three institutes (Institute of Biology, Ecology and Agricultural Technologies; Institute of Physical Education, Sport and Tourism; Institute of Education and Psychology). The length of the pilot stage of the project was from September 2017 up to December 2017 (one semester).

### 2.2. Teachers

Six teachers of the English Language Department of the Petrozavodsk State University participated in the action research project. Their teaching experiences ranged from 10 up to 30 years.

## 3. Methods and Phases of the Research Project

The action research project focused on restructuring the university FL curriculum intended for bachelor's degree pre-service non-language teachers. The project was carried out on three phases: the preliminary phase, the implementation phase, and the final phase. Both qualitative and quantitative methods were applied in the process of data collection. Parts of the results were statistically analyzed. A wide range of the methods applied in the project allowed us to build a sufficient database for the detailed description of the context and for the identification of the main directions of the new, restructured university FL curriculum.

During the preliminary phase of the research, the data were obtained: (a) through interviews and discussions with university FL teachers who later participated in the project; (b) through the results of placement tests that all the student participants (first-year students) did; (c) through the analysis of FL curricula of other Russian universities located in the adjacent regions; (d) through comparing some aspects of the related data in Russian and Finnish universities; (e) through conducting a questionnaire with open-ended questions among the student participants in the project (152 respondents), which was designed to find out the students' attitudes to learning FLs in the previous years of education;

(f) through the needs analysis of school subject teachers concerning the use of an FL in their jobs; (g) through gathering data related to the university context of FL teaching and learning, including the time allocated to the subject and FL materials intended for the target group of students; (h) through the analysis of the existing approaches to FL university teaching. The overview of the data gathered in the initial phase (see Part 4 of the article) allowed us to diagnose the most crucial drawbacks and problem areas of teaching and learning FLs in the context under study, to articulate research questions for the project, and to outline a few solutions to consider in the study.

In the process of conducting the project (the implementation phase), the data were obtained through: (a) the teachers' classroom observations; (b) the teachers' forum in the social network VKontakte, where the teacher participants exchanged their field notes and suggested new ideas on how to reorganize the teaching process; (c) through brief checks of the student current progress, which allowed for keeping track of their learning and introducing quick changes in the course of classroom activities, if required. The immediate collaborative character of the study in this phase facilitated the further refinement of both classroom as well as independent home activities of the learners (see Part 5 of the article).

In the final phase of the project implementation, all the students did the same FL test, which included reading, speaking, and writing sections. The assignments were designed in such a way that every student could demonstrate his/her FL proficiency level in the given situations. There were no multiple-choice tasks, which tend to limit the students' reactions within the imposed contexts. The participants were expected to express their own ideas in the FL using the vocabulary and the sentence structures they had learned in the course of the project and which they considered appropriate for their self-expression. Quantitative methods were used to analyze the test results—for example, the shift in the number of the students with different levels of the FL competency was proved statistically through Chi-square test (see Part 6 of the article).

The factors in the study that were not changed as compared with the previous course were the number of the classroom hours (90 min per week) and the length of the semester in formal university FL education (first semester: four months, eighteen 90 minute classes).

The factors that were reexamined in the course of the project were: (a) the lowest acceptable FL proficiency level that the students had to achieve to pass their end of year FL exam; (b) the content of the course; (c) classroom techniques; (d) organization of the students' self-directed studies.

Therefore, it was a complex team study conducted in a real-life educational context, which provided preliminary findings for further restructuring the university FL curriculum intended for pre-service subject teachers.

## 4. The Results of the Initial Phase of the Project

### 4.1. The Study of the Target Learners

Having analyzed the data that describe the target students, we identified a few critical features of the learners, which should be taken into consideration while organizing their university FL education.

Firstly, the initial level of FL competency of the target first-year students does not meet the requirements of the Russian school standards in FL education, nor does it live up to the university teachers' expectations. At the beginning of the semester, a placement test was conducted in every student group. The format, the materials, and the assessment criteria for the university placement test were similar to the FL standardized tests used in Russia for secondary school graduates. However, in contrast to the FL school graduation test, it consisted of three parts—reading, use of the FL, and writing. In Russia, every school graduate must achieve level B1 in compliance with the Common European Framework Reference for Languages (CEFR) where each level is clearly defined. The CEFR provides detailed descriptors of the FL competences and sub-competences specific for each level [1]. The assessment criteria were developed on the basis of the Federal (National) State Secondary Education Standard (FSSES-FGOS). The students who did not meet the FL proficiency level required by the FSSES

were ranked among those with a low proficiency level, those who met the language level partly were among those with a medium proficiency level, and, finally, the students who met the requirements presented were among those with a high FL proficiency level.

The placement test revealed very low results; 80% of the students demonstrated levels A1-A1+; 9%–level A2; 11%–level B1. A considerable variety of the levels of different FL skills and sub-skills demonstrated by separate students was also evident in every academic group (for example, B1 in reading and A1 in the use of the FL). The use of FL grammar appeared to be the weakest feature in practically all the tests.

Secondly, a questionnaire filled out by the students showed that only 30% of them were interested in learning FLs at school. The majority of the respondents (73%) were aware of the importance of FL education in modern life and would like to enhance their initial level. They mentioned the following reasons for why they may need the FL: participation in student exchange programs (11%) and teacher research conferences (6%), reading scientific sources (17%), and doing leisure activities (91%); the latter was the most frequently mentioned factor.

If we consider the university student needs for FL communicative competence (Table 1), we can identify the following ones: participation in exchange programs or international studies, reading research books or articles for graduation papers (thesis), or doing leisure activities. Meanwhile, the analysis of 100 graduation papers (thesis) of the bachelor's degree graduates in Education of the Petrozavodsk State University revealed that only five graduates used sources in English, which demonstrated their low language level and lack of motivation to use materials in FLs in their leaning activities.

**Table 1.** Motivation for learning foreign languages (FL) at school.

| Reasons for Learning a FL | Interest in Learning FLs | Awareness of the Importance of FLs | Participation in Exchange Programs | Research Conferences | Reading Scientific Sources | Doing Leisure Activities |
|---|---|---|---|---|---|---|
| Number of students | 30% | 73% | 11% | 6% | 17% | 91% |

Similarly, the statistics of the number of the PetrSU students in Education taking part in international exchange programs indicate that they account for less than 1% of all those involved.

The same trend was noticed by the Finnish Centre for International Mobility (CIMO) [2]. Students in Education accounted only for 2% of international students in 2015. For the next two years, there was even a 6% decline of the Finnish students studying abroad in this field [3]. Part of the reason is also related to the insufficient FL level. However, in Finland, the number of students in Education increases significantly in Master's programs, while a similar growth in the number of PetrSU Master's degree students in Education taking part in exchange programs is unlikely to occur, as the English Language course is limited to the first year of university education.

As the data above indicate, the overwhelming majority of the students in the target group have very vague motivational orientations for learning FLs. What makes matters worse is the fact that only few of the students from the target group make sufficient efforts for the realization of these needs.

Following the guidelines of the Common European Framework of Reference for Languages: Learning, Teaching, Assessment Companion Volume With New Descriptors [1] (p. 26), we made an attempt to comply with the needs analysis oriented towards real-life tasks, "working backwards from what the users/learners need to be able to do in the language". We found out that it is practically impossible to design a clear and realistic FL needs profile of subject school teachers, because FLs are not used by them in their jobs. In reality, no professional tasks require FL competency. We did not manage to identify relevant FL needs except for a limited number of cases when subject teachers participated in school exchange programs, which was very rare.

Therefore, we can conclude that university FL teachers have to rely mostly on the students' interest in using FLs in their leisure activities. Another more important direction of boosting the students' motivations is trying to involve them in some international programs as a close realistic target to strive for. Due to the absence of clear needs and opportunities to use FLs in their jobs, the pre-service as well as the in-service non-language subject teachers are different from other groups of university FL learners, majoring, for example, in business, medicine, or tourism. It is obvious that they need a different FL course with a focus on general FLs.

*4.2. The Study of the Teaching and Learning Context of University Foreign Language Education of the Target Group of Students*

The study of the context of university FL education included the analysis of the factors related to the classroom time, the length of the course, the number of the students in academic groups, and the availability of publications intended for the target group.

The PetrSU curriculum for the target group of students allocates only one class (90 min) per week to FL education during the first year of university studies. After the final examination in the FL (the end of the second semester), students are not offered any language courses. A review of the programs of a few Russian universities in the north-western regions also shows that they allocate about 60–80 teaching classroom hours to FLs.

Another challenge in the context of the project is related to the number of mixed-ability students in the classroom (about 25–30 students), which requires teachers to use a variety of materials and well-thought-out techniques.

Meanwhile, the analysis of student books in professional university FLs revealed a significant abundance of learning materials for students in Business, Economics, Computer Science, and Engineering learning English, for example (Table 2). Meanwhile, students in Education cannot be provided with any learning materials of the leading publishing companies, as these materials do not exist in Russia.

**Table 2.** Number of course books in professional FLs of the leading publishing companies in different fields.

| | Business and Economics, Law | Computer Science, Engineering, Construction, Oil Industry, Oil and Gas | Science, Biology, Geography | Aviation | Medicine Nursing | Education |
|---|---|---|---|---|---|---|
| Cambridge University Press | 13 | 5 | 1 | 1 | 2 | 0 |
| Macmillan | 8 | | | 3 | | 0 |
| Oxford University Press | 10 | 10 | 1 | 2 | 5 | 0 |

The factors above reveal major problems related to the university FL education of the target group of students. One of them concerns learning time. The amount of FL classroom time is obviously insufficient for improving the students' FL proficiency levels. The Introductory Guide to the Common European Framework of Reference (CEFR) for English Language Teachers [4] (p. 4) estimates that learners typically need approximately 350–400 guided learning hours to progress to level B1 and even more time to achieve higher levels. Given that each classroom consists of big groups of students with a great variety of FL proficiency levels and there are no properly developed and tested materials and techniques intended for the student target group, the conclusion is obvious—the learning outcomes of every classroom depend on the quality of the curriculum and on the teacher's professional competence.

*4.3. The Approaches and Methods of University Foreign Language Education*

Currently, there are several models of teaching FLs to non-language university students, which vary according to the aims and the content of the graduates' professional profiles.

- *English or Language for Specific Purposes* (ESP/LSP) refers to teaching the English language to university students or people already in employment, with reference to the particular vocabulary and skills they need on their jobs. As with any language taught for specific purposes, a course of ESP will focus on one occupation or profession, such as Technical English, Scientific English, English for Medical Professionals, etc. [5]. The ideas of T. Hutchinson and A. Waters (ESP founders) were developed in various educational models, all of which shift their focus to professional and vocational learning FLs, meeting the needs of specialists through professional content, vocabulary, and tasks aimed at coping with professional challenges by means of an FL [5,6].
- *English for Academic Purposes* (EAP) focuses instruction on skills required to perform in an English-speaking academic context across core subject areas generally encountered in a university setting [7].
- *Business English* focuses on vocabulary and topics used in the worlds of business and on the communication skills used in the workplace, such as presentations, negotiations, meetings, small talk, socializing, correspondence, and report writing. As Evan Frendo notes, " . . . business English is an umbrella term for a mixture of general everyday English, general business English and ESP" [8].
- *Vocationally Oriented Language Learning* model (VOLL) centers around learning a language, skills, and subject content for work and life [9,10]; "Unlike LSP or ESP, VOLL does not have highly specialized professional contexts as a focus of instruction" [10] (p. 66). It emphasizes interpersonal communication in a FL, including situations in professional contexts.
- *Content and Language Integrated Learning* (CLIL) [11–18] refers to teaching subjects such as science, history, and geography to students through a foreign language. CLIL is sometimes referred to as having "four Cs" as components—content, communication, cognition, and culture [14]. This is a useful description because the integration of content, communication, cognition, and culture is one way to define teaching aims and learning outcomes.
- *Productive FL Learning* [19–21] refers to the "orientation of a holistic educational process to a professional foreign language product, self-education and self-actualization of the learner's personality in result of his/her active learning experience" [20] (p. 320). This approach focuses on the integral development of both values and a set of competencies, including professional ones. Thus, teaching and learning FLs is supposed to contribute to the whole personal development of the student.

The ESP as well as the CLIL models are intended for business, financial, engineering, and tourism students who are aware of the clear perspectives of the FL proficiency for their future work life. In the field of pre-service non-language teacher education, there is no certainty concerning the importance of learning an FL for the future professional needs. It inevitably reduces the level of the students' motivation.

The CLIL model cannot be used in this context either for the following reasons. Firstly, the Content and Language Integrated Learning creates a challenge for FL teachers who have to know at least the basics in the particular subject content. In Russian reality, university FL teachers are not proficient in professional subjects. As a result, they cannot select and teach professional content independently from subject teachers, while subject teachers are not able to deliver lectures in an FL. Secondly, in the Russian context, the FL course is taught to pre-service teachers only during the first year of University studies along with other courses of general knowledge. The professional subjects are mainly taught later. Therefore, professional texts and situations that are used in the FL classroom at this time usually contain superficial or trivial information. Thus, because of the students' low proficiency levels as

well as of the absence of the subject knowledge, reading and speaking in an FL may easily turn into primitive quasi professional communication. Finally, the CLIL approach at the level of higher education implies the use of at least an intermediate level of FL, which is a challenge for the majority of non-linguistic students.

According to the FSUES in Russia, all the bachelor's degree students are supposed to achieve proficiency level B2 in General English. In addition, they are expected to develop FL professional competences, which will allow them to participate in oral and written communication around professional topics. It implies that, for example, pre-service teachers of biology will have to discuss and read English texts both on biology and pedagogy. Therefore, the amount of the vocabulary sub-skills as well as the quality of all the skills that the target students will have to develop enormously increases. However, those young people who enter the university with levels A1–A2 are not able to cope with the task.

In addition to different models of university FL teaching, we would like to mention internationalization of higher education, which has become a popular trend all over the world with the rationale of developing cross-cultural understanding and cooperation among people based on the intercultural communication competence [22–27]. Practical internationalization of higher education through teaching professional courses in an FL and students' participation in international projects, international mobility, internship in multicultural environment, and distance learning courses in FLs is a powerful impetus for non-linguistic students to learn an FL more actively [24]. We share the idea that "internationalization must not only be outward-looking, but also inward-looking" [26] (p. 331). In our view, it is an effective way to boost students' needs awareness of learning FLs, especially in the context of unclear motivational orientations.

Thus, we can conclude that none of the FL teaching and learning models listed above are a clear fit in the case of university FL education of the target group of students.

### 4.4. Results of the Initial Phase Analysis

Summing up, we can identify the constraints that affect the characteristics of the university FL curricula intended for the target group of the students in the project:

- Initial low FL proficiency level of the majority of the target first-year students in Education;
- A wide range of levels of different FL skills (listening, speaking, reading, and writing) and sub-skills (vocabulary and grammar);
- Overcrowded mixed-ability groups of learners seated in the same classroom;
- Unclear student needs and vague perspectives of the future applicability of the FL;
- Unreasonably short classroom time allocated by some universities to FL education of future non-language teachers;
- Absence of good-quality, time-tested teaching and learning FL materials intended for the FL education of the target group of students;
- Contradictions between the students' awareness of the importance of FL competencies in modern society and their unwillingness (or inability) to use FL skills in real life situations for student exchange programs or for writing their graduation papers.

Therefore, it is no wonder that there is a gap between the desired outcomes defined in the FSUES and those achieved in the university FL education of the target group of students. The most sensible and realistic ideas for the design of an effective university FL curriculum in the given context are those with an emphasis on learning general FLs for life (VOLL), on the simultaneous all-around personal as well as communicative development of the target students (Productive FL Learning Model), along with measures for internationalization of their education. To enhance the students' interest in FL learning and to boost their efforts, it would also make sense to involve them in all kinds of real life international activities where FLs are required.

## 5. The Results of the Implementation Phase of the Research Project

The implementation phase of the project consisted of identifying the objectives, the content, and the techniques of the university FL curriculum intended for the pre-service non-language subject teachers. It also included a practical part when the proposed ideas were tested in actual teaching. The section below presents the summary of the constituents of the curriculum.

### 5.1. Objectives of the Foreign Language Course for Pre-Service Subject Teachers

Due to the prevailing low FL proficiency levels of the majority of the first-year students, the objectives of the FL course were described as ranging from the lowest acceptable level (the FL functional literacy) up to level B2 on the Common European Framework of Reference for Languages (CEFR), though the highest outcomes remained open depending on the student aspirations and efforts s/he was ready to make in FL learning. The FL functional literacy was defined as the ability to use the General FL for fulfilling common, everyday functions fluently in listening, speaking, reading, and writing, being an agent of simple FL communication. Thus, the FL functional literacy served as the lowest acceptable level for assessment and as a solid basis for further growth in the FL competency.

The professional focus of the suggested FL course was not so much on the subject matter related to the particular student's major. The emphasis was placed on the development of a few teaching generic skills that could be effectively nurtured in the given setting: (a) information processing in the course of FL oral and written interaction (sharing, inquiring, analyzing, summarizing, and evaluating); (b) FL public speaking skills while presenting information to a group of people; (c) peer-tutoring, peer-control, and peer-assessment.

Special attention was paid to the development of the efficient FL learning strategies, which could help students cope with the existing problems and shape the necessary functional instruments for further self-regulated FL acquisition. Setting realistic goals, finding appropriate tools and resources, as well as monitoring the process and the results of independent home work were the key skills that were fostered for further FL learning.

### 5.2. Content of the Foreign Language Course for Pre-Service Subject Teachers

The content of the FL curriculum in the university education of pre-service subject teachers was built on the idea of the priority of the General FL used for performing communicative receptive and productive language activities. We proceeded from the assumption that professional language is rather a combination of General FL in a professional environment and the use of specific vocabulary [28]. Therefore, the most frequent General English words as well as international words (such as "smart", "enter", "assist", etc.) constituted the core vocabulary of the course. The acquisition of these international words did not require special cognitive efforts on the part of learners. Another important observation was that the FL vocabulary of the students with different proficiency levels could vary a great deal. However, the core vocabulary was a must for every student to be used.

In language practice tasks, we also emphasized general FL grammar material (Simple Tenses, Degrees of Comparison, the use of the verb "to be", etc.). We assumed that the development of sustainable FL sub-skills was an essential prerequisite for effective communication in any context, because students felt more confident in cases where they had a good command of grammar and vocabulary even though they were not very extensive.

The majority of the topics for discussion and reading (subject matter) were presented in a spiral format. The emphasis in the subject matter was placed not on specific subject knowledge but on the peculiar features of the teacher's profession. For example, during the first semester, the students discussed four basic issues (Daily Routine and Lifestyles; Work Life; Free Time Activities; Building Relationships). Each issue was approached from three different angles—from a general perspective, from the student's perspective, and from the teacher's perspective. Thus, in each module, we presented the following sub-topics: (1) A Daily Routine and Lifestyle of an Ordinary Person; (2) Student's Daily

Routine; (3) Teacher's Daily Routine; or (1) Building Relationships with the People Around; (2) Building Relationships with Fellow Students and Teachers; (3) Building Relationships with Colleagues, Students, and Their Parents, etc.

The specificity of the subject matter could vary depending on the students' particular field of study. Thus, future PE teachers could discuss special diets for sportsmen as well as the careers of prominent athletes while, for example, future elementary school teachers could read and talk about younger children's popular pastimes and games. Within this approach, the same topic required the use of approximately the same language units but in slightly different situations. On the one hand, it enhanced the students' language sub-skills. On the other, it developed the ability to transfer words and sentence structures into slightly new contexts.

The main communicative functions that the students practiced in the tasks were descriptions, asking for information, analyzing the collected information (comparing, classifying), summarizing, providing arguments for and against, and evaluating. These were the key functions that allowed the students to build well-grounded opinions for oral and written discourse.

The information for discussions was collected by the students either from texts (written or oral) or from other students (in pairs, groups, or mingles) or was derived from the students' personal experiences.

In the classroom, the learners were asked to share with their fellow students the information they had acquired in their independent studies. Therefore, they had to learn how to select the most important facts and then how to present them to a group or to individual students. That is why the student independent activities were organized in such a way as to promote the development of both self-regulation skills in independent work and teacher's generic skills in information processing and public speaking. Part of the content in the course was related to peer-tutoring, peer-control, and peer-assessment when, in addition to sharing information, the students were asked to fulfill a few common teacher's functions.

*5.3. Teaching Techniques and Student Activities in the Foreign Language Course for Pre-Service Subject Teachers*

In designing the classroom and the independent work techniques, we proceeded from the following assumptions related to the organizational aspects of teaching and learning:

- *"Diversity of mixed ability students/diversity of materials and tasks"* [29] as the fundamental principle of teaching and learning management. This principle implies tailoring materials and tasks to the abilities of the students involved. A set of diverse tasks, texts for listening and reading, problems, and project assignments are grouped around one particular topic and offered to different students as their homework assignments. In the classroom, it entails varied forms of student interaction while exchanging information (or checking or discussing, peer-tutoring or peer-evaluation, etc.);
- The individual learning trajectory [30] for every student as a way to follow in self-study. These trajectories are elaborated in line with the students' personal needs, aspirations, as well as learning problems. They are used to organize both the student's online and traditional home studies. The tasks and the materials are differentiated into three levels (A, B, and C) with regard to their difficulty. Every student can choose which level s/he needs to master at the moment and then progress to a higher one. Students can choose those assignments and as many of them as they find adequate to meet their individual goals or to effectively solve specific learning problems. The teacher informs them where the assignments are available and which of them are a required minimum to be done. Students are encouraged to practice more in case they set higher achievement goals for themselves. The results of their individual homework assignments are presented in students' portfolios. This approach to managing the students' independent work contributes to the development of self-regulated learning strategies;
- *Multifunctional chains of tasks*, which allows students to master the same language units and subject matter from different perspectives. A multifunctional chain involves fulfilling a variety of

tasks based on the same material. Each new task in the chain changes the learner's focus. Each following assignment relies on what has been practiced in the previous one and appears to be more complicated. For example, based on the video they watch, students first list the events and the characters, then they give more detailed descriptions of each, after which they classify the information (or compare facts that they learned). Finally, they come up with their opinions, providing arguments from the video. This chain is based on one material and fulfills varied functions in the development of the FL competence.

- *Priority of varied ways of student classroom interaction* (pairs, groups, mingles, and circle work as a way of peer communication, peer teaching, and peer control), which enhances student involvement in communicative FL practice and their speaking time in the classroom. This active involvement noticeably improves student fluency.

The interrelated implementation of the listed guidelines promotes the student agency both in FL learning as well as in FL communication.

## 6. The Results of the Final Phase of the Project

The final phase of the project consisted of doing the final FL test, summarizing the students' independent work results during the semester, and analyzing the outcomes. The analysis included a comparison of the results from both the placement test and the final test and their subsequent discussion.

In the final test, the students first read a text about stress in general (its causes, signs, and ways to cope with it). For checking reading comprehension, they had to do a multiple-choice assignment. Then, every student discussed stress in student life with the teacher (speaking). The last part was writing an opinion essay about stress in teachers' lives. The topic had not been discussed during the classes, but all the topics covered before could be incorporated into each of the tasks. Thus, the students were expected to demonstrate a flexible use of the FL dealing with a new subject matter. Unlike the placement test, the final test did not check the use of the FL in a separate assignment. The level of the students' abilities to use the FL was determined based on their accurate and fluent application of vocabulary and grammar while speaking and writing. The objective of the final test was to evaluate the end of the semester FL proficiency level that every student participant achieved. Three levels were identified: low level (A1–A1+), medium level (A2–A2+), or high level (B1 and higher). To make the descriptors and the criteria for each level easily applicable, we tried to avoid long and ambiguous descriptions. On the whole, the criteria used in both tests were close in meaning and emphasized the ability to use an FL (grammar and vocabulary) for performing communicative tasks.

The description of the low level (A1): students demonstrate (a) a general comprehension of the text (reading); (b) understanding of the teacher's simple questions; (c) ability to give brief answers; (d) ability to express their opinion in simple words; (e) ability to correctly use limited FL vocabulary and grammar.

The description of the medium level (A2): students demonstrate (a) a general comprehension and a partly detailed comprehension of the text (reading); (b) understanding of all the teacher's questions; (c) ability to give more detailed answers; (d) ability to express their opinions in simple words, providing one or two arguments/examples without extended detail; (e) ability to correctly use FL vocabulary and grammar.

The description of the high level (B1): students demonstrate (a) a detailed comprehension of the text (reading); (b) complete understanding of the teacher's questions, dealing with specific issues of the problem under discussion; (c) ability to give detailed answers; (d) ability to express their opinions, providing extended arguments/examples; (e) ability to correctly use extended FL vocabulary and grammar.

The efficiency of the developed curriculum for a university FL course was assessed based on the comparison of the students' results before (IN) and after (OUT) the implementation phase according to the following criteria:

*Communication criterion* (Co IN/Co OUT) reveals the ability to achieve the aim of communication in accordance with the task. It also takes into account the number of utterances produced by every student in speaking and writing.

Evaluation based on the communicative criterion:

- For reading:

  - General comprehension: maximum 8 points (level A2—functional literacy);
  - Detailed comprehension: maximum 12 points (level B1).

- For speaking and writing:

  - Ability to give general facts and ideas on the topic in a brief text (up to 8–10 sentences): maximum 6 points (level A2—functional literacy);
  - Ability to give more detailed oral or written answers on the topic (up to 11–15 sentences): maximum 6 points (level A2+);
  - Ability to provide extensive arguments/examples (up to 16–20 sentences): maximum 8 points (level B1).

*Vocabulary criterion* (Voc IN/Voc OUT) shows diversity (range) of the vocabulary units used in speaking and writing.

Evaluation based on the vocabulary criterion:

- Ability to use simple FL words repeating them occasionally: maximum 6 points (level A2—functional literacy);
- Ability to use FL vocabulary correctly avoiding repetition: maximum 8 points (level A2+);
- Ability to use diverse FL vocabulary correctly avoiding repetition: maximum 8 points (level B1).

*Grammar criterion* (Gr IN/Gr OUT) focuses on grammatical accuracy in speaking and writing.
Evaluation based on the grammar criterion:

- Ability to use simple FL grammar correctly—not more than four errors: maximum 6 points (level A2—functional literacy);
- Ability to use FL grammar correctly—not more than two errors: maximum 8 points (level A2+);
- Ability to use diverse FL grammar correctly—not more than two errors: maximum 8 points (level B1).

*Functional learning criterion* (FL IN/FL OUT) includes regularity of the student self-study, development of the language portfolio, and active participation and initiative during the class and in self-study. In their initial interviews, students were asked to assess their readiness and their ability to fulfill self-regulated work. In the final phase, students' FL learning efforts were assessed by the teachers. They were granted scores in accordance with the following scale:

- Portfolio: coverage of every topic discussed during the semester—5 points for each;
- Every paper corrected after the teacher's checking—2 points for each;
- Classroom participation—5 points for each class;
- The prescribed minimum of the self-study assignments—5 points for each;
- Every extra self-study assignment—2 points for each.

Evaluation:

- 240 points and more—"excellent" (high level);
- 239–200 points—"good" (high level);

- 199–170 points—"satisfactory" (medium level);
- Less than 170 points—poor (low level).

The detailed results of both the placement test (Table 3) and the final test (Table 4) are presented in the tables below.

**Table 3.** Results before the main phase of the project shown in student percentages.

| Criterion | Low Level | Medium Level | High Level |
|---|---|---|---|
| vocabulary | 65% | 22% | 13% |
| grammar | 87% | 3% | 10% |
| communication | 68% | 22% | 10% |
| functional learning | 42% | 32% | 26% |

**Table 4.** Results after the main phase of the project shown in student percentages.

| Criterion | Low Level | Medium Level | High Level |
|---|---|---|---|
| vocabulary | 12% | 14% | 74% |
| grammar | 24% | 54% | 22% |
| communication | 12% | 24% | 64% |
| functional learning | 6% | 24% | 70% |

Consolidated results before (IN) and after (OUT) the implementation phase of the project are presented in Table 5.

**Table 5.** Results before and after the main phase of the project shown in student percentages.

| | Voc IN | Gr IN | Co IN | FL IN | Voc OUT | GR OUT | Co OUT | FL OUT |
|---|---|---|---|---|---|---|---|---|
| Low | 65% | 87% | 68% | 42% | 12% | 24% | 12% | 6% |
| Medium | 22% | 3% | 22% | 32% | 14% | 54% | 24% | 24% |
| High | 13% | 10% | 10% | 26% | 74% | 22% | 64% | 70% |

* Voc: vocabulary criterion; Gr: grammar criterion; Co: communication criterion; FL: functional learning criterion.

We can also display the final shift of the low, the medium, and the high students' FL proficiency levels in graphs (Figure 1).

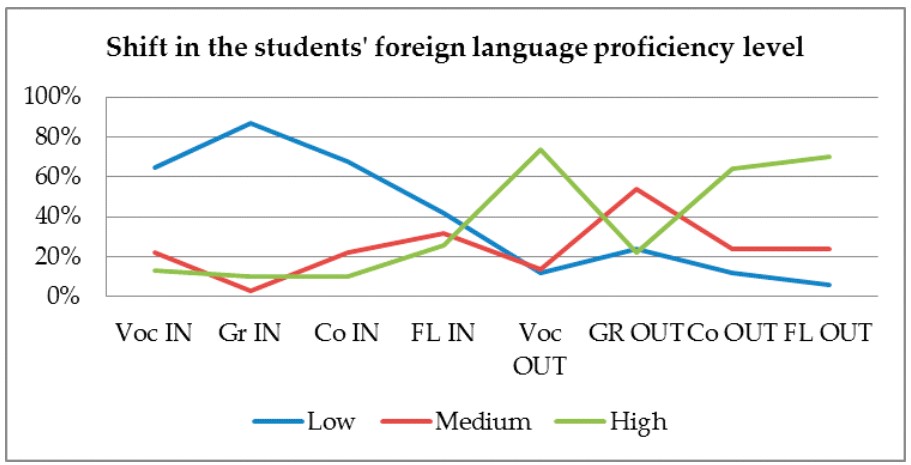

**Figure 1.** The students' foreign language proficiency levels.

The shift in the number of the students with low level of the FL competency was proved statistically through Chi-square test. The results of the test on each criterion are presented in the following tables (Tables 6–9).

**Table 6.** Results of Chi-square statistic on vocabulary criterion.

|  | Voc IN | Voc OUT | Row Totals |
|---|---|---|---|
| Number of students with low level of FL competency | 99 (58.50) [28.04] | 18 (58.50) [28.04] | 117 |
| Number of students with medium level of FL competency | 33 (27.00) [1.33] | 21 (27.00) [1.33] | 54 |
| Number of students with high level of FL competency | 20 (66.50) [32.52] | 113 (66.50) [32.52] | 133 |
| **Column Totals** | 152 | 152 | **304 (Grand Total)** |

The chi-square statistic is 123.7737. The *p*-value is < 0.00001. The result is significant at $p < 0.05$.

**Table 7.** Results of Chi-square statistic on grammar criterion.

|  | Gr IN | Gr OUT | Row Totals |
|---|---|---|---|
| Number of students with low level of FL competency | 132 (84.00) [27.43] | 36 (84.00) [27.43] | 168 |
| Number of students with medium level of FL competency | 5 (44.00) [34.57] | 83 (44.00) [34.57] | 88 |
| Number of students with high level of FL competency | 15 (24.00) [3.38] | 33 (24.00) [3.38] | 48 |
| **Column Totals** | 152 | 152 | **304 (Grand Total)** |

The chi-square statistic is 130.7435. The *p*-value is < 0.00001. The result is significant at $p < 0.05$.

**Table 8.** Results of Chi-square statistic on communication criterion.

|  | Co IN | Co OUT | Row Totals |
|---|---|---|---|
| Number of students with low level of FL competency | 104 (61.00) [30.31] | 18 (61.00) [30.31] | 122 |
| Number of students with medium level of FL competency | 33 (34.50) [0.07] | 36 (34.50) [0.07] | 69 |
| Number of students with high level of FL competency | 15 (56.50) [30.48] | 98 (56.50) [30.48] | 113 |
| **Column Totals** | 152 | 152 | **304 (Grand Total)** |

The chi-square statistic is 121.718. The *p*-value is < 0.00001. The result is significant at $p < 0.05$.

**Table 9.** Results of Chi-square statistic on functional learning criterion.

|  | FL IN | FL OUT | Row Totals |
|---|---|---|---|
| Number of students with low level of FL competency | 63 (36.00) [20.25] | 9 (36.00) [20.25] | 72 |
| Number of students with medium level of FL competency | 49 (42.50) [0.99] | 36 (42.50) [0.99] | 85 |
| Number of students with high level of FL competency | 40 (73.50) [15.27] | 107 (73.50) [15.27] | 147 |
| **Column Totals** | 152 | 152 | **304 (Grand Total)** |

The chi-square statistic is 73.0257. The *p*-value is < 0.00001. The result is significant at $p < 0.05$.

## 7. Discussion

Analyzing the statistical results of the project, it is evident that, although there seemed to be a stable growth in all the aspects of the student FL proficiency, the most unsatisfactory results were observed in the use of grammar. Both the in- and the out-tests revealed that FL grammar sub-skills caused problems for accurate and fluent self-expression for the majority of the target students. This

could be accounted for by the abstract character of grammar rules and considerable differences between Russian and English grammar as well as.

The best results were displayed in the use of vocabulary, which we attributed to a number of special measures in selecting and practicing FL vocabulary. Good results were also achieved within the communication criterion. In our view, this confirms the benefits of the interaction techniques widely used in the classroom. While preparing their home assignments, students had to engage in active reading and listening to be able to share the information in the classroom later. This made them more focused on the communicative tasks.

The results relating to the students' achievements in developing their functional learning strategies techniques were really valuable for their further growth given the limited time of the university FL education. The teachers monitored the students' self-regulated studies on the basis of their reports, portfolios, and presentations of their home assignments to the class and also checked their written papers and online work results. It was possible to see which tasks or texts the students chose, when and how often they interacted with the learning materials, and how well they were progressing. We can definitely state that the majority were learning to design and follow their individual learning trajectory, choosing wisely what they really needed for better outcomes. For the students with the initial low level, it was a good chance to achieve the level of functional literacy and pass the exam. For more successful students, it gave a real opportunity to make further progress. In any case, the application of the trajectory proved to be beneficial for organizing self-regulated learning.

These results indicate that 12% of the participants did not make any noticeable progress during the semester. Nevertheless, judging by the results, half of them improved their self-study strategies, which allowed for predicting better outcomes during the second semester. The analysis shows that from 60% up to 70% of those who demonstrated improvements within the functional learning criterion also achieved better results in other areas (vocabulary, grammar, and communication). Thus, the importance of the interrelated development of the FL competences and of learning strategies was evident.

The students engaged in the project turned out to be more active in accessing different forms of extra-curricular activities than those who were not. Twice as many of them (compared to the number of the participants in the previous years) took part in the PetrSU Student Research Conference where they made presentations in English; 40% of all the presentations at the conference were made by the students of the Institute of Physical Education, Sport and Tourism, where the placement test had revealed unsatisfactory levels of the FL competence, and 70% of the first-year non-language bachelor's degree students in Education voluntarily took part in the large scale PetrSU English Language Olympic Marathon. In the interviews with the teacher staff, the students underlined that they would willingly participate in different FL leisure activities such as fan clubs of English language songs and movies, joint events with the students of the Foreign Language Institute of the PetrSU (concerts, performances, and debates), as well as short-term language camps together with international students. They also suggested that specialized FL courses devoted to reading and discussing scientific literature would be very useful in the process of preparing their graduation papers. We completely support their suggestions. It would make sense to incorporate short FL courses where the target students could activate their FL competences watching videos of their major subject lessons in schools in the English language countries or reading and discussing articles on their major subjects from different sources.

Summarizing our answers to the research questions posed at the beginning of the article, we conclude that the key feature of the university FL curriculum is diversity. Given that there are mixed ability students with vague needs and motives of FL learning seated in the same classroom, the curriculum suggests a range of open-ended objectives of FL education, a variety of content for acquisition, and a diversity of tasks that can be adjusted to different students. The outcomes of every student depend on the effort and the time s/he is ready to invest in learning FLs, although each learner is provided with obvious opportunities to acquire sufficient General FL skills for further learning.

## 8. Conclusions

The project implemented by the FL teachers of the PetrSU confirms our preliminary proposals regarding new directions for teaching and learning FLs in the context of pre-service non-language subject teacher education. All the factors discussed in the present paper place the task of looking for effective ways of inspiring the target students to achieve sufficient and realistic FL proficiency levels in their university education among the most crucial ones. The data we presented in the article constitute only a very modest beginning toward meeting this challenge. They outline the pathway for further, more detailed research.

**Author Contributions:** Conceptualization, E.B. and M.S.; Methodology, E.B.; Investigation, M.S.; Data Curation, E.B. and M.S.; Validation, M.S.; Formal Analysis, E.B.; Writing—original draft preparation, E.B. and M.S.; Software, M.S.; Visualization, M.S.; Writing—review and editing, E.B.

**Funding:** This research received no external funding.

**Conflicts of Interest:** The authors declare no conflict of interest.

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
