# Peer review of "A University Foreign Language Curriculum for Pre-Service Non-Language Subject Teacher Education"

_education, doi:10.3390/educsci9030163_

Round 1
Reviewer 1 Report
The topic is very interesting and the paper has a great potential for publishing. It needs to be improved methodologically. The whole research is design is quite poor. The research methodology part does not specify what kind of research is carried out, whether it is a quasi-experiment, action research, observation, interview. In different parts of the text we learn a little about which research methods are used for collecting data. Reseach sample is not defined and reseach aims do not correspond with the research question. There are no results from the observations and interviews. Authors state results only from the tests and these are repeated several times in various forms. The research question is not answered.
Author Response
Thank you very much for your time and efforts to give comments on our article.
Taking into account all the recommendations we have:
- changed the title of the paper to a more specific one “The University Foreign Language Curriculum for Pre-Service Non-Language Subject Teacher Education”
- restructured the article in a different way (Methods, Participants, Results of the Initial Phase of the Project, Results of the Main Phase of the Research Project, Final Results of the Project, Discussion, and Conclusions)
- defined the research (see lines 98-99)
- specified the methods used (see Section 2 “Methods and Phases of the Research Project”)
- provided more detailed and precise description of the methods and the process of the study
- clarified the research question (lines 40-42), the objectives of the study (lines 45-51), and the ways of collecting data and statistics (Section 2)
- introduced the results from students’ interviews.
Reviewer 2 Report
see attachment

Author Response
Thank you very much for your time and efforts to give comments on our article.
Having taken into consideration all the comments, we have introduced some major changes.
1. The title of the paper has been changed to a more specific one “The University Foreign Language Curriculum for Pre-Service Non-Language Subject Teacher Education”.
2. The paper has been rewritten and restructured in a more traditional format (Methods, Participants, Results of the Initial Phase of the Project, Results of the Main Phase of the Research Project, Final Results of the Project, Discussion, Conclusions)
3. The abstract has been rewritten in a more precise way.
4. The participants have been described in a more specific way.
5. In the paragraph about needs analysis we tried to highlight the contradiction between the students` needs as they see them (based on the questionnaires) and their actions in learning FLs (based on the data concerning their graduation papers and participation in exchange programmes)..
6. The message of the paragraph on the commonly used approaches to university FL education is to prove that, due to the absence of clear needs and opportunities to use FLs on their jobs, the pre-service as well as the in-service non-language subject teachers are different from those learners who fall into other groups of learners. Therefore, they need a different FL course with a focus on a general FL.
7. The “uniqueness” of the curriculum suggested in the article is that its key idea is diversity. Given that there are mixed ability students with vague needs and motives of FL learning seated in the same classroom, the curriculum suggests a range of open-ended goals, a variety of content for acquisition, and a diversity of tasks. The outcomes of every student depend on the efforts and time s/he is ready to invest (the individual learning trajectory).
8. The description of the constituents of the curriculum has been changed. We included more details and examples concerning the goals, the content, and the techniques.
9. We have tried to illustrate and specify the teaching techniques and student activities
10. The criteria have been explained and the evaluation process has been described in more detail.
11. The results have been reported in a different way, using Chi-square test. All the calculations have been provided. Figure 1b has been removed as it did not add any information to the paper.
12. We also added some detail on self-regulated learning. The concept of the individual learning trajectory is at the root of the self-study in this case. Students choose which proficiency level they want to achieve, which materials and tasks to do and how much time to devote to learning the language.
Reviewer 3 Report
The paper presents an original approach which is of interest for Language teaching in Russia. it has merit and needs to be supported. However there are major amendments required before publication. I have uploaded the original document with some suggestions and comments for the author to consider. In summary:
need to add more clarity to the contextual information and announce the structure of the paper in the Introduction section
Stylistic changes required throughout as it is awkward at times. I suggest to seek the service of a professional editor if possible
The section Participants and Context does not hang well together. I suggest to separate the context section into "General educational context" and " T&L context of FLE" perhaps?
The project design and implementation section needs an introductory sentence to foreground what the section is about.
In this section the author gives some result before explaining the design of the project. it is confusing. Similarly, is the placement test part of the implementation of the project, or is it applied to all students? The author needs to reorganise some of the content in this section for clarification purposes.

Author Response
Thank you for uploading the document with your comments and remarks, we appreciate it a lot.
Taking into consideration your recommendations, we have introduced significant changes in the structure of the paper and its design. The title of the paper has been changed to a more specific one “The University Foreign Language Curriculum for Pre-Service Non-Language Subject Teacher Education”.
The paper has been rewritten and restructured in a more traditional format (Methods, Participants, Results of the Initial Phase of the Project, Results of the Main Phase of the Research Project, Final Results of the Project, Discussion, and Conclusions). The paper has been reorganized and more specific detail has been provided. Generally, we have tried to make the paper more specific and the description of our study clearer.
Round 2
Reviewer 1 Report
The article has been improved. However, there are still missing research aims and corresponding research questions, which should be answered at the end of the paper.
Author Response
Thank you very much for the tıme you spent on our manuscrıpt and for your comments.
We have trıed to take ınto account all your comments and we have ıntroduced the research aıms and corresponding research questions, which we answered at the end of the paper
Reviewer 2 Report
The authors have done a great job reframing their study. However, some of the flaws in the first draft, especially being more specific about measures, criteria and data analyses (see my comments in the document), are still present in the revised version. So I would encourage the authors to provide where possible more detailed information on method. Finally, the manuscript would benefit from a thorough proofreading.

Author Response
Thank you very much for the tıme you spent on our manuscrıpt and for your comments.
We have trıed to take ınto account all your comments and we have
added the ınformatıon concernıng the methods used (sectıon 2)
clarıfıed the way the data were collectad and analyzed (section 2 )
restructured the paper: put the descrıptıon of the partıcıpants fırst and methods second (sectıons 2 and 3)
provıded more detaıled ınformatıon on the placememt test used ın the course of the project (sectıon 4.1)
provıded more ınformatıon on the assessment crıterıa used (sectıon 6)
Reviewer 3 Report
I am satisfied with the major amendments to this paper. I am attaching the revised version with a few stylistic suggestions for the author (see yellow highlights with attached notes) to improve the overall readability before final publication.

Author Response
Thank you very much for the tıme you spent on our manuscrıpt and for your comments.
We have trıed to take ınto account all your comments and we are sure the ıntroduced changes improve the overall readability of the paper.